# Interrelation between α-Cardiac Actin Treadmilling and Myocardin-Related Transcription Factor-A Nuclear Shuttling in Cardiomyocytes

**DOI:** 10.3390/ijms23137394

**Published:** 2022-07-02

**Authors:** Mark-Alexander Gorey, Mathias Mericskay, Zhenlin Li, Jean-François Decaux

**Affiliations:** 1Institut de Biologie Paris-Seine (IBPS), CNRS UMR 8256, INSERM ERL U1164, Biological Adaptation and Ageing, Sorbonne Université, 75005 Paris, France; alex0gorey@gmail.com (M.-A.G.); jean-francois.decaux@inserm.fr (J.-F.D.); 2INSERM UMR-S 1180, Signalling and Cardiovascular Pathophysiology, Université Paris-Saclay, 92296 Châtenay-Malabry, France; mathias.mericskay@inserm.fr

**Keywords:** MRTFA, nucleocytoplasmic shuttle, α-cardiac actin, SRF, RhoA/ROCK signaling pathway, polymerization, PI3K/Akt signaling pathway

## Abstract

Myocardin-related transcription factors (MRTFs) play a central role in the regulation of actin expression and cytoskeletal dynamics that are controlled by Rho GTPases. SRF is a ubiquitous transcription factor strongly expressed in muscular tissues. The depletion of SRF in the adult mouse heart leads to severe dilated cardiomyopathy associated with the down-regulation of target genes encoding sarcomeric proteins including α-cardiac actin. The regulatory triad, composed of SRF, its cofactor MRTFA and actin, plays a major role in the coordination of the nuclear transcriptional response to adapt actin filament dynamics associated with changes in cell shape, and contractile and migratory activities. Most of the knowledge on the regulation of the SRF–MRTF–Actin axis has been obtained in non-muscle cells with α-actin and smooth muscle cells with α-smooth actin. Here, we visualized for the first time by a time-lapse video, the nucleocytoplasmic shuttling of MRTFA induced by serum or pro-hypertrophic agonists such as angiotensin II, phenylephrine and endothelin-1, using an MRTFA-GFP adenovirus in cultures of neonatal rat cardiomyocytes. We showed that an inhibitor of the RhoA/ROCK signaling pathway leads to an α-cardiac actin polymerization disruption and inhibition of MRTFA nucleocytoplasmic shuttling. Moreover, inhibition of the PI3K/Akt signaling pathway also prevents the entry of MRTFA into the nuclei. Our findings point out a central role of the SRF–MRTFA–actin axis in cardiac remodeling.

## 1. Introduction

The heart is capable of structural and functional remodeling in response to environmental stimuli that can induce the growth or the shrinkage of the organ. In this process, the serum response factor (SRF) and the myocardin-related transcription factor-A (MRTFA), a coactivator of SRF, can play a major role. A number of studies in mouse models have shown that the modulations of the activity of SRF causes alterations in cardiac function [1,2]. From a physiological point of view, the role of SRF in the regulatory triad including MRTFA, actin and SRF itself has not been established in cardiomyocytes, and the precise mechanism and visualization of MRTFA translocation remains clearly unresolved. This type of shuttling, in relation to non-muscle β-actin, has been described in cultured fibroblast cells [3,4].

SRF and coactivators of the myocardin family are the main regulators of the transcription of contractile, energy transfer and cytoskeletal genes [1,5,6,7]. The myocardin-related transcription factors including MRTFA (also known as MAL or MKL1) and MRTFB (also known as MKL2) are widely expressed in numerous tissues [8] and are also essential in cardiomyocytes [6]. Studies in non-cardiac cells have shown that actin filament dynamics are linked tightly with the activity of SRF and that the polymerization of globular actin liberates MRTF cofactors, allowing their nuclear translocation. Once in the nucleus, interaction between MRTF cofactors and SRF enhances the transcription of various genes (SRF-dependent transcription) [3,4,9,10,11]. In addition, MRTFA-mediated gene expression has been shown to be regulated by the activation of the RhoA/ROCK signaling pathway [10,12,13]. The PI3K/Akt signaling pathway is known to play an important role in cardiac function, and a potential relation between PI3K, RhoA/ROCK and actin organization has been shown in various cell types [14,15,16], suggesting that it could affect the MRTFA nuclear shuttling.

In the present study, by using an MRTFA-GFP fused protein produced by a recombinant adenovirus and the time-lapse live cell imaging, the nucleocytoplasmic shuttle kinetics of MRTFA and actin dynamics were analyzed in primary cultures of neonatal rat cardiomyocytes. Our results have shown for the first time that stress induction by serum or pro-hypertrophic agonists leads to the nucleocytoplasmic shuttling of MRTFA-GFP linked to an increase in filamentous actin dynamics. However, this shuttle can be blocked by the ROCK inhibitor Y-27632. Interestingly, we reported a potential interaction between RhoA/ROCK and PI3K/Akt signaling pathways in this process.

## 2. Results

### 2.1. Detection of MRTFA-GFP in Cardiomyocytes but Not in Fibroblasts

Neonatal rat cardiomyocytes (NRCs) were isolated with a Percoll purification procedure. This approach eliminates the majority of cardiac fibroblasts. Firstly, we assessed the purity of isolated primary NRCs. The immunostaining of vimentin (Figure 1A,B) and desmin (Figure 1D,E) antibodies was used to visualize fibroblasts and cardiomyocytes, respectively. Our data showed that fibroblast level accounts for 20.8% and cardiomyocytes for 79.2% of a total cell (Figure 1C). Moreover, we observed that the MRTFA-GFP fusion protein was only found in cardiomyocytes stained by the desmin antibody and never in cardiac fibroblasts stained by the vimentin antibody (Figure 1A,B,D–F), showing the preference of the adenovirus MRTFA-GFP to infect the NRCs. This specificity facilitates our study in the NRCs with this adenovirus.

After adenovirus infection, the localization of the MRTFA-GFP fusion protein was analyzed in cardiomyocytes that were cultured in a serum-free DMEM medium for 24 h. We showed that the MRTFA-GFP protein was mainly located in the cytoplasm of cardiac cells (Figure 1F). Primary cultures of NRC were analysed under a spinning disk microscopy time-lapse over a period of 60 min with intervals of 5 min between pictures. To ascertain the distribution of MRTFA-GFP between the cytoplasm and the nucleus, the average mean intensity of the MRTFA-GFP signal was quantified in the nucleus (N) and an area of the cytoplasm (C) using imageJ software. The ratio of the nuclear to cytosol signal was determined and the cells were arbitrarily divided in three categories: N/C < 1: predominantly cytosolic localization of MRTFA-GFP; 1 < N/C < 1.5: significant nuclear localization with remnant MRTFA-GFP in cytosol and N/C > 1.5: predominantly nuclear localization of MRTFA-GFP. We showed that in these baseline serum-starved conditions, MRTFA is located in the cytoplasm in 65–67% of the cells, or both in the cytoplasm and the nucleus in 22–25% of cells and is predominantly nuclear in less than 3% of cardiomyocytes (Figure 1G). The distribution of the MRTFA-GFP protein in the three categories was not changed between the time points 0 and 60 min during the experiments (Figure 1G).

### 2.2. Effect of Serum, Endothelin-1, Phenylephrine or Angiotensin II on MRTFA Nucleocytoplasmic Shuttling in Cardiac Cells

We assessed the kinetics of the nucleocytoplasmic shuttle of MRTFA-GFP using serum stimulation in NRCs. Serum stimulation was performed by adding 20% serum (13.5% HS plus 6.5% FCS) in DMEM medium. The live cell imaging was performed at 5 min intervals over a 60 min period by using inverted spinning disk microscopy. MRTFA-GFP was undetectable in the nucleus at time 0 (Figure 2A,B). The kinetics of MRTFA-GFP translocation showed an immediate nuclear accumulation, most of the translocation occurred within the first 5 min and the maximal translocation was observed at 20 min and remained stable in the nucleus over a period of 60 min (Figure 2A and Appendix A). One hour after serum induction, 70% of MRTFA-GFP was localized in the nucleus, and a complete loss of the GFP signal was observed in the cytoplasm (Figure 2B). This experiment shows that MRTFA nucleocytoplasmic shuttling is fast after serum addition in cultured cardiac cells.

These cells are also responsive to pro-hypertrophic agonists such as phenylephrine (PE), an adrenaline analog that stimulates α-adrenergic receptors, or endothelin-1 (ET-1) and angiotensin II (Ang II) that are peptide hormones triggering cardiomyocyte hypertrophy [17,18,19]. We treated NRCs with ET-1, PE or Ang II at different time points ranging from 5 min to 3 h to determine the kinetics of MRTFA-GFP nuclear translocation. Before the addition of agonists, cells were cultured in a serum-deprived medium for 24 h. Whatever the agonist used, the maximum MRTFA-GFP translocation was obtained between 40 and 60 min after addition of the hormones and continued gently until 3 h to reach more than 63%, 74% or 84% of nuclear MRTFA-GFP with ET-1, Ang II or PE, respectively (Figure 2C–E and Figure 3C, Appendix A).

### 2.3. Effect of Y-27632, a ROCK Inhibitor, on MRTFA Shuttling in Cultured Cardiomyocytes

To examine the impact of actin dynamics on the MRTFA shuttle in cardiomyocytes, we used an inhibitor of the RhoA signaling pathway. Y-27632 [(+)-(*R*)-trans-4-(1-aminoethyl)-N-(4-pyridyl) cyclohex-anecarboxamide dihydrochloride] is an inhibitor of Rho-associated coiled-coil forming protein serine/threonine kinase (ROCK) that is found downstream in the RhoA pathway and is involved in the polymerization of actin through the phosphorylation and activation of Lin-11, Isl-1 and Mec-3 kinase 1 (LIMK1), which itself phosphorylates and inhibits profilin actin depolymerization activity [20,21]. Twenty min after the addition of Y-27632 into NRCs cultured in the serum-free medium, we observed an MRTFA-GFP perinuclear accumulation as “a ring”, which was maintained until 60 min (Figure 3B). This perinuclear accumulation trend was also seen in basal conditions in the absence of Y-27632, although it appeared to be more diffuse (Figure 3, compare A to B). At 180 min, a fraction of the MRTFA-GFP protein returned to the cytoplasmic area (Figure 3B). The addition of Ang II led to a progressive nuclear accumulation of MRTFA between 60 to 180 min (Figure 3C). Then we tested if Y-27632 could inhibit the nuclear translocation of MRTFA. Cells were placed in serum-deprived DMEM medium for 24 h, then treated for 30 min with Y-27632 before the addition of Ang II. We obtained a similar profile as previously shown for the inhibitor alone with a blockage at the stage of perinuclear ring accumulation of MRTFA-GFP and finally diffusion back to the cytoplasm (Figure 3D). These data show that the RhoA/ROCK signaling pathway plays a crucial role in the nucleocytoplasmic shuttling of MRTFA downstream of the Ang II peptide hormone in cardiomyocytes. 

### 2.4. Filamentous Actin Dynamics in Relation to MRTFA Shuttling in Cultured Cardiomyocytes

To assess the relationship between MRTFA-GFP shuttling and cellular filamentous actin dynamics, we co-infected NRCs with MRTFA-GFP alone or in the presence of Life-Act-RFP, a marker to visualize F-actin [22]. Cells were treated with Ang II for 180 min then fixed with PFA for immunofluorescence analysis. Under control serum-free conditions, a specific antibody was used to visualize the α-cardiac actin isoform in cardiomyocytes infected by adenovirus MRTFA-GFP. We showed that MRTFA-GFP localized to the cytoplasm and showed a partial striated pattern, although it was only rarely overlapping with α-cardiac actin filaments in line with the idea that cytoplasmic MRTFA is mostly bound to the G-actin pool and not to the F-actin (Figure 4A). Ang II treatment led to MRTFA-GFP translocation towards the nucleus and the increased striation of α-cardiac actin (Figure 4B). To obtain a brighter more complete staining of the actin networks present in the cardiomyocytes, we used the Life-Act-RFP probe, a 17-amino acid peptide derived from a yeast protein fused to red fluorescent protein (RFP) that stained F-actin structures in living or fixed cells. Life-Act-RFP was clearly incorporated in the striated sarcomeric actin pool without evidence of an additional non-sarcomeric network (Figure 4C). These data suggest that Life-Act-RFP mainly targets α-cardiac actin, which is the major isoform of actin expressed in cardiomyocytes. In the presence of Y-27632, the actin filament’s striation pattern was only partially reduced and the MRTFA-GFP signal remained perinuclear (Figure 4D). The addition of Ang II, 30 min after the inhibitor, resulted in the MRTFA-GFP signal remaining mainly cytoplasmic with a perinuclear ring accumulation. Under these conditions of pretreatment with Y-27632, the increase in actin striation induced by Ang II was much less pronounced than in the absence of the inhibitor (Figure 4, compare E to C).

### 2.5. Effect of Wortmannin and LY294002, Two PI3K Inhibitors, on MRTFA Shuttling in Cultured Cardiomyocytes

To establish a potential involvement of the PI3K signaling pathway on the MRTFA shuttle in cardiomyocytes, we used wortmannin and LY294002, two distinct inhibitors of PI3K. Wortmannin acts by an irreversible inactivation of PI3K by covalent modification of the catalytic subunit [23,24], while LY294002 reversibly inhibits PI3K by competing with ATP for the active site of catalytic subunit p110 [24,25]. Similar to the results shown in Figure 2, more than 70% of MRTFA-GFP is localized in the nucleus 60 min after serum induction (Figure 5B) compared with the basal conditions where more than 95% of MRTFA-GFP is detectable in the cytoplasm (Figure 5A). Cardiac cells were placed in a serum-deprived medium for 24 h then treated for 30 min with wortmannin or LY 294002 before the addition of the serum. A similar distribution profile of MRTFA-GFP in the cytoplasm of the cardiomyocytes treated with the inhibitors alone was observed as that observed in Figure 5A (data not shown). We observed a blockage of MRTFA-GFP in the cytoplasm area 60 min after the addition of the serum (Figure 5C,D). In addition, we tested the effects of a pro-hypertrophic agonist (PE) in the conditions of PI3K inhibition. The experiment was carried out under the same conditions as the serum, just with it replaced by PE. After 180 min of PE stimulation, we confirmed the nuclear translocation of MRTFA-GFP compared with the basal conditions (Figure 6, compare A to B). The addition of wortmannin or LY294002 leads to a blockage of MRTFA-GFP in the cytosol despite the presence of PE (Figure 6C,D). These data show that the PI3K signaling pathway is involved in the MRTFA shuttling process.

## 3. Discussion

Here we report for the first time that actin stress fiber induction by serum or by pro-hypertrophic agonists leads to the increased nuclear localization of MRTFA-GFP in cardiac cells in a primary culture. This process can be blocked by the ROCK inhibitor Y-27632 but also by the PI3K inhibitors, wortmannin and LY294002, suggesting a crucial impact of the RhoA/ROCK and PI3K signaling pathways on the MRTFA nucleocytoplasmic shuttle. 

Cardiomyocytes, placed in a serum-deprived medium, showed a cytoplasmic localization of the MRTFA-GFP fused protein. G-actin is exported out of the nucleus and accumulates in the cytoplasm, effectively sequestering MRTFA outside of the nucleus since MRTFA strongly interacts through its N-terminal RPEL domain to G-actin only, as previously described in NIH3T3 fibroblasts [4,10,26]. Upon serum stimulation or pro-hypertrophic agonist treatments, we observed that MRTFA-GFP accumulated in the cardiomyocytes nuclei in line with the stimulation by these effectors of G-actin’s incorporation into F-actin. With the serum, MRTFA-GFP’s nuclear accumulation was effective in a few minutes and almost complete in less than one hour where it remained for a significant period (more than three hours), indicating a rapid incorporation of G-actin into F-actin. This process was comparatively slower with the pro-hypertrophic agonists, suggesting that the serum could contain a broader spectrum of active molecules acting through parallel pathways onto actin polymerization. Once in the nucleus, the MRTFA interacts with SRF as homo- or hetero dimers to drive SRF-dependent gene transcription [11]. A hundred genes have been shown to be transcriptionally regulated by SRF over the last 30 years [27,28,29,30]. The persistence over at least 3 h of MAL/MRTF relocalization to the cardiomyocytes’ nuclei upon serum stimulation was also described in fibroblast cells [10]. The importin α/β1 heterodimer regulates the nuclear import of MRTFA in response to the activation of the RhoA signaling pathway in cultured vascular smooth muscle cells [31]. 

In our cultured cardiac cells, we demonstrated that the stimulation of nuclear MRTFA-GFP transport by Ang II led to a rapid increase in striated actin polymerization labeled by the α-cardiac actin specific antibody. Moreover, Life-Act, which can bind to all isoforms of actin as long as they are forming F-actin did not reveal a significant pool of non-striated actin (Figure 4). Hence, α-cardiac actin represents a major partner for MRTFA and is also the main isoform of actin involved in the polymerization process in cardiac cells. The second interesting aspect was to determine the impact of the RhoA/ROCK signaling pathway in this process. Using the ROCK inhibitor, Y-27632, we showed that the translocation of MRTFA-GFP to the nucleus was unable to take place. In this condition of ROCK inhibition, we observed the perinuclear accumulation of MRTFA that could be due to a concentration of monomeric actin unable to polymerize in that region of the cell, therefore disabling the nuclear translocation of MRTFA. In addition, our data indicate that there is a relationship between α-cardiac actin, MRTFA and SRF following the activation of the RhoA/ROCK signaling pathway in cultures of neonatal rat cardiomyocytes. The inability of MRTFA to translocate to the nucleus in Y-27632-treated cells, even after pro-hypertrophic induction, would confirm that one of the major pathways induced by hypertrophic agonists in cardiomyocytes is indeed the RhoA/ROCK signaling pathway through actin polymerization. This suggests that the repression of the RhoA/ROCK signaling pathway could be used as a means to repress the overstimulation of the MRTFA-SRF complex leading to pro-hypertrophic genes expression.

Results from previous studies have pointed out an important role played by the PI3K/Akt signaling pathway in cardiac function and in cardiac protection [32,33,34,35]. It has been reported that transgenic PI3K mice were resistant to cardiac dysfunction induced by pressure overload [32]. Moreover, the overexpression of PI3K in mice with dilated cardiomyopathy delayed the onset of heart failure with an increase in lifespan [33]. More recently, connections between PI3K, RhoA/ROCK and actin organization have been described in various cell types [14,15,16]. In our study, we demonstrate that the inhibition of the PI3K/Akt signaling pathway by specific inhibitors prevents the MRTFA-GFP entry into the nuclei after stimulation by the serum or a hypertrophic agonist such as PE in cardiac cells in the primary culture. It is interesting to note that the inhibition of the RhoA/ROCK or PI3K/Akt signaling pathways leads to the blockage of MRTFA-GFP into the nucleus. This could suggest that an activation of PI3K signaling selectively modulates the actin dynamic and MRTFA activity through the RhoA/ROCK activation pathway. In addition, this process promotes MRTFA-SRF association leading to an increase in SRF-dependent gene expression. Recent data have shown that SRF regulates craniofacial development by PDGF signaling through the recruitment of MRTF cofactors [36]. Further investigations will be required to identify the factors responsible to determine whether the effects of PI3K signaling in the cascade result in MRTFA nuclear shuttling in cardiomyocytes or if PI3K is only required as a permissive factor but additional pathways such as RhoA/ROCK are actively required. 

In conclusion, this study showed and visualized, for the first time, that the nucleocytoplasmic shuttle of MRTFA (using an MRTFA-GFP fused protein) is linked to an increase in filamentous cardiac actin dynamics in the primary culture of cardiomyocytes. In addition, this MRTFA transport is intimately dependent on the RhoA/ROCK signaling pathway, and the PI3K/Akt signaling pathway is also involved in this process. Based on these data, future directions will attempt to further explore the dynamic changes of α-cardiac actin under pro-hypertrophic agonists and also biochemical stress. Using cellular and molecular methods of protein analysis and a combination of inhibitors will help to further breakdown the role of specific pathways involved in actin dynamics in cardiac cells.

## 4. Materials and Methods

### 4.1. Isolation of Neonatal Rat Cardiomyocytes

Isolation of neonatal rat cardiomyocytes was performed as previously described [5]. NRCs were seeded at a density 7.5 × 10^4^ cells/cm^2^ in Ibidi plates (Cliniscience-80446) previously coated with 10 μg/mL of laminin (BD Biosciences) cultured at 37 °C in 5% CO_2_. Using Ibidi 4 well-chambered coverslip allows the direct visualization under the exact same conditions of control conditions while inducing a variety of stresses simultaneously. They allow the elimination of the meniscus due to the chambered coverslips, enabling the direct visualization of cardiomyocyte cultures, live cell microscopy and immunofluorescent staining of cells.

### 4.2. Culture Conditions

Cells were cultured in DMEM containing 5.5 mM glucose and *GlutaMAX™* supplemented with 5% FCS, 10% HS and 50 U/mL penicillin and 50 µg/mL streptomycin at 37 °C in a humidified atmosphere of 5% CO_2_. At day 2, cells were infected with adenovirus MRTFA-GFP (a gift of Dr A. Sotiropoulos, Institut Cochin, Paris, France) or with adenovirus rAV^CMV^-Life-Act-TagRFP (Ibidi, Cliniscience) at 50 MOI/cell overnight [22]. For co-infection experiments, we first performed Ad-Life-Act-Red infection at 25 MOI/cell for 6 h followed by adenovirus MRTFA infection at 25 MOI/cell in refreshed growth medium. At day 4, cells were washed twice for 10 min with PBS (pH 6.5), then once for 10 min with PBS (pH 7.5) before cultured in serum-deprived medium for 24 h. Then, stress induction was performed and cells were observed using inverted spinning disk microscopy with a strictly regulated environment of 5% CO_2_ and at 37 °C. Serum stimulation was performed using medium with 20% serum (13.5% HS plus 6.5% FCS). The treatment with pro-hypertrophic agonists was performed with Ang II, PE or ET-1 at 100 nM. For the experiments including the inhibitor Y-27632 (Sigma), the ROCK inhibitor at 10 μM was added 30 min before the treatment of the appropriate pro-hypertrophic agonist at 100 nM. Inverted spinning disk microscopy was used for live cell imaging at a magnification of 40 taking pictures at 5 min intervals over varying periods of time ranging from 60 to 210 min. For the experiments including the inhibitors LY294002 (Sigma) or wortmannin (Sigma) at 30 μM and 100 nM, respectively, the PI3K inhibitors were added 30 min before the stimulation of 20% serum or of PE at 100 nM. The cardiac cells were observed using inverted spinning disk microscopy. 

### 4.3. Fluorescence Microscopy

Immunofluorescent staining was performed as previously described [1] from cell cultures fixed with 3.7% (*v*/*v*) formaldehyde for 10 min then washed with PBS (pH 7.5). Immunofluorescence analysis involved the following primary antibodies: anti-vimentin (1:500, Progen, Heidelberg, Germany), anti-desmin (1:500, Genemed, South San Francisco, CA, USA) and anti-α-cardiac actin (1:500, American Research Products, Massachusetts, USA). A Cy3-coupled secondary antibody (1:2500, Sigma, St Quentin Fallavier, France) was used to detect the expression of these three proteins. Nuclei were counterstained with DAPI dye (1:10,000, ThermoScientific, Waltham, MA, USA). For the analysis, an inversed confocal microscopy located at the platform “Cellular imagery of IBPS Sorbonne Universite” was used.

### 4.4. Quantification of MRTFA-GFP Distribution

To ascertain the distribution of MRTFA-GFP between the cytoplasm and the nucleus, the average mean intensity of MRTFA-GFP signal was quantified in the nucleus (N) and an area of the cytoplasm (C) using imageJ software. The ratio of the nuclear to cytosol signal was calculated. The cells were arbitrarily divided in 3 categories: N/C < 1: predominantly cytosolic localization of MRTFA-GFP, 1 < N/C < 1.5: significant nuclear localization with remnant MRTFA-GFP in cytosol, N/C > 1.5: predominantly nuclear localization.

### 4.5. Statistical Analyses

All the results were expressed as means ± SEM The significance of differences between means was assessed with both Student’s *t*-test and Mann–Whitney’s test for non-gaussian data. *p*-value of < 0.05 was considered to be statistically significant.

## Figures and Tables

**Figure 1 ijms-23-07394-f001:**
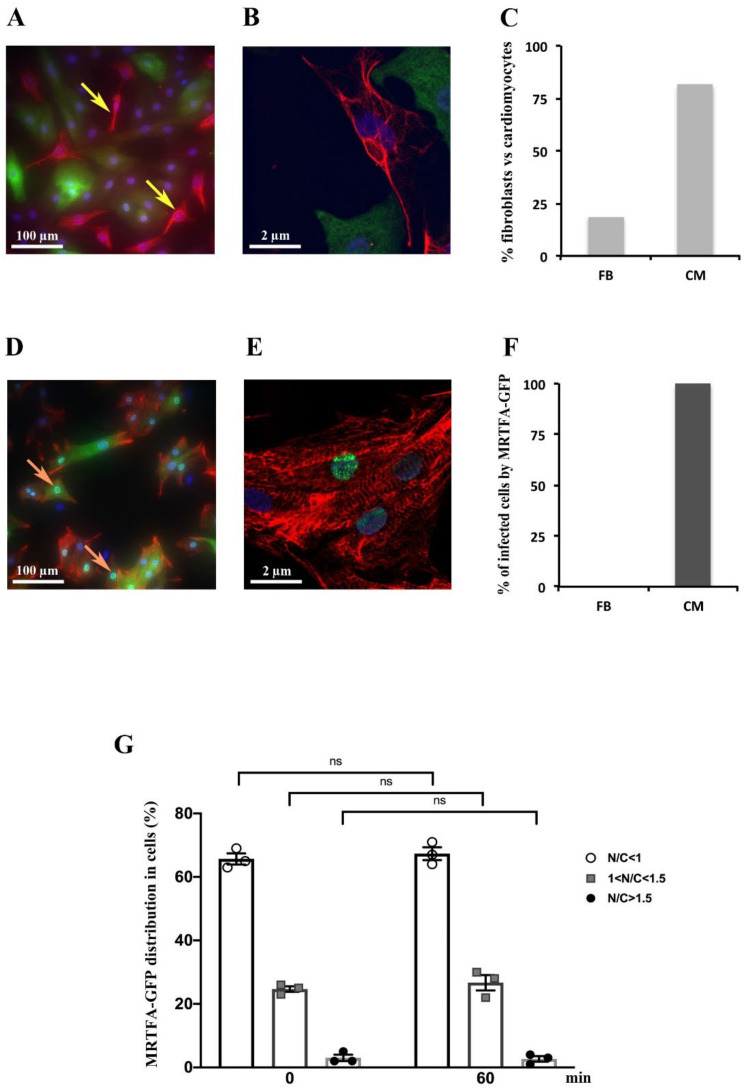
Expression and localization of MRTFA-GFP fusion protein on neonatal rat cardiomyocytes. Cells were infected by adenovirus MRTFA-GFP at 50 MOI and starved in serum free DMEM medium for 24 h. Serum stimulation was performed by adding 20% serum (13.5% HS plus 6.5% FCS) to medium during 60 min in (**A**–**F**). (**A**,**B**) Immunofluorescence labeling of vimentin (red), and MRTFA-GFP (green). The yellow arrows indicate fibroblasts. (**C**,**D**) Immunofluorescence labeling of desmin (red), The orange arrows indicate cardiomyocytes including MRTFA-GFP localized in the nucleus (green). (**E**) Histogram representing percentage of cardiomyocytes (CM) and fibroblasts (FB) in cultured cells. (**F**) Histogram of percentage of CM and FB having MRTFA-GFP protein. (**G**) Percentage of three categories of MRTFA-GFP distribution in NRCs under basal conditions (serum free) at the beginning (0 min) and the end (60 min) of experiment. ns: non-significant. These data are representative of 3 independent experiments, *n* = X − Y (range) cells per analysis.

**Figure 2 ijms-23-07394-f002:**
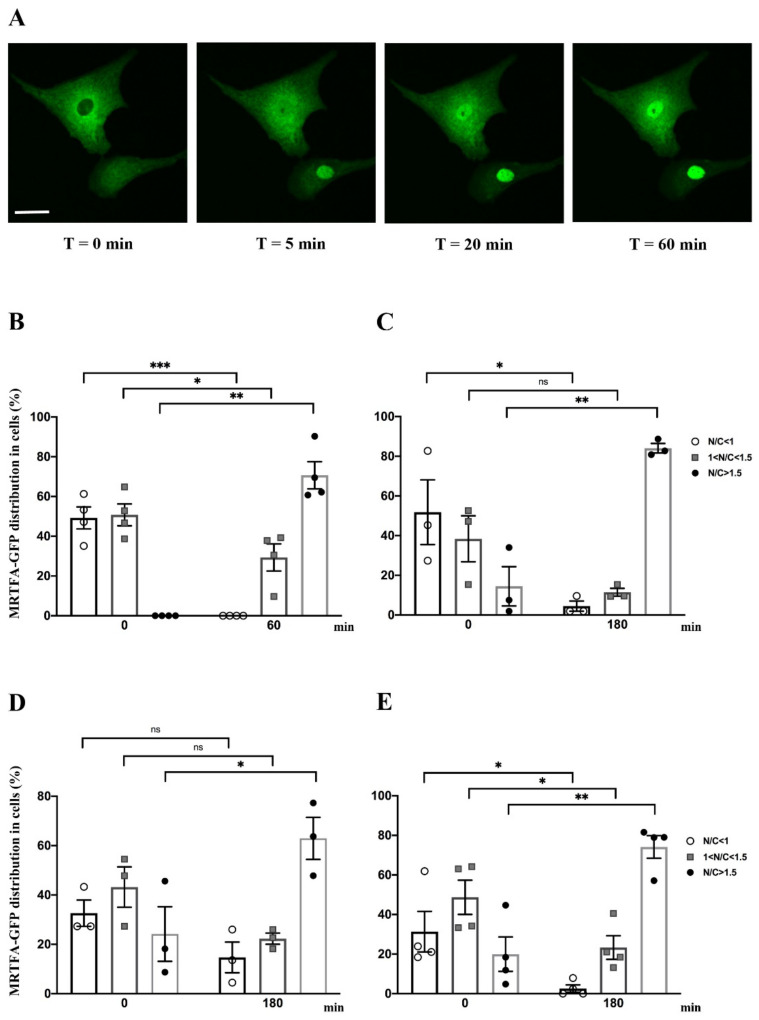
Impact of serum, phenylephrine, endothelin-1 and angiotensin II on MRTFA-GFP shuttling. NRCs were starved in serum-deprived DMEM medium for 24 h. Serum stress was conducted by adding serum to medium to obtain 20% final concentration (13.5% HS plus 6.5% FCS) at time 0 then followed at 60 min. NRCs were also treated with pro-hypertrophic agonists (Ang II, PE or ET-1) at 100 nM at time 0 then followed at 180 min. (**A**) Fluorescence image of MRTFA-GFP protein in NRCs that were starved and then stimulated by serum at time 0, 5, 20 and 60 min. Scale bar: 20 µm. Percentage of three MRTFA-GFP distribution categories in NRCs; (**B**) serum stimulation conditions at 0 and 60 min; (**C**) treatment with PE at 0 and 180 min; (**D**) treatment with ET-1 at 0 and 180 min; (**E**) treatment with Ang II at 0 and 180 min. Data are mean ± SEM. *, ** and *** indicate significant difference, respectively, at *p* < 0.05, *p* < 0.01 and *p* < 0.001 vs time 0, ns: non-significant. These data are representative of 3 to 4 independent experiments.

**Figure 3 ijms-23-07394-f003:**
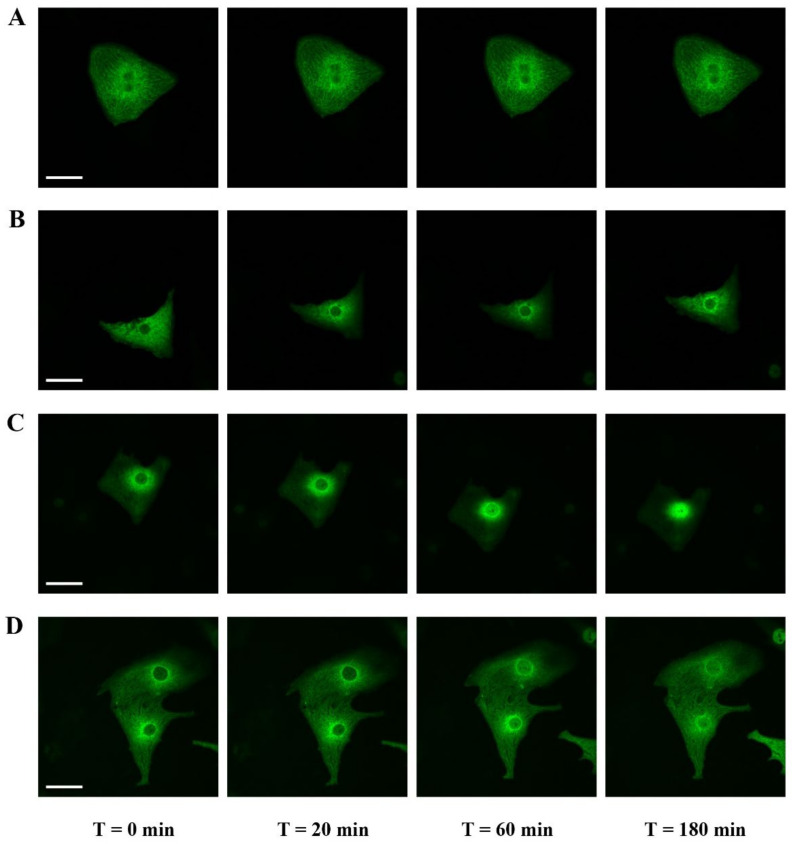
Effects of ROCK inhibitor (Y-27632) on the MRTFA-GFP shuttle. NRCs were infected by adenovirus MRTFA-GFP at 50 MOI, starved in serum-deprived DMEM medium for 24 h before stress induction. Fluorescence image of MRTFA-GFP, (**A**) basal conditions (serum free); (**B**) basal conditions and treatment with Y-27632 (10 µM); (**C**) treatment with Ang II (100 nM); (**D**) treatment with Ang II (100 nM) + Y-27632 (10 µM). Scale bar: 20 µm. These data are representative of 3 independent experiments.

**Figure 4 ijms-23-07394-f004:**
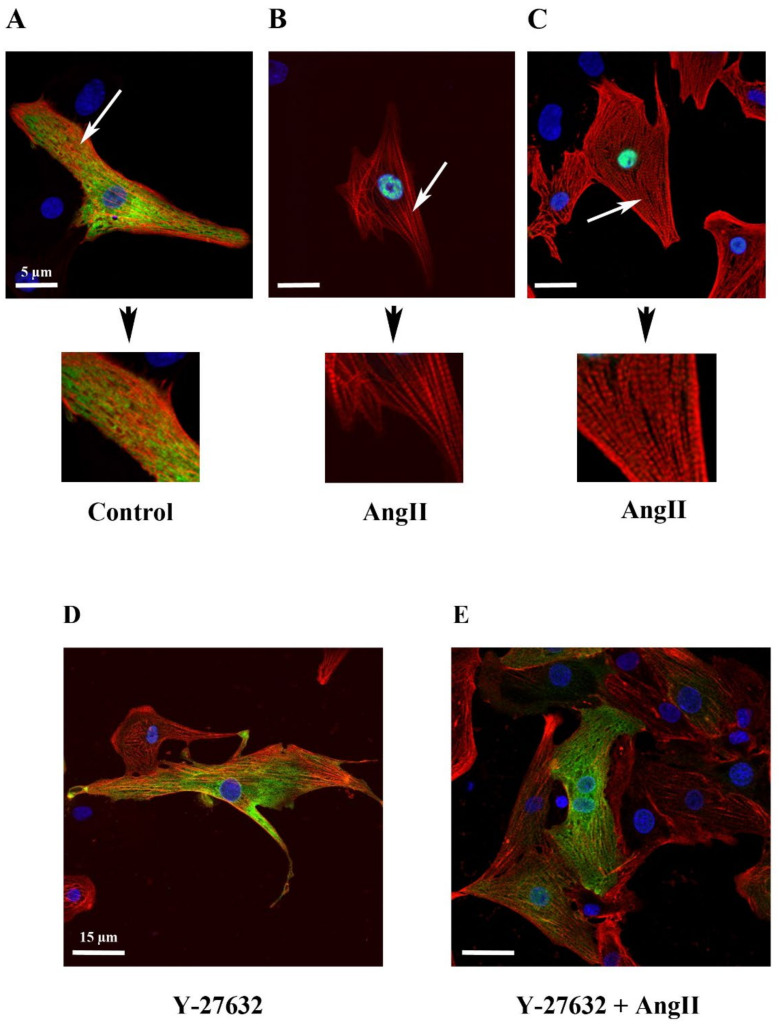
Impact of ROCK inhibitor (Y-27632) on actin network in relation to MRTFA-GFP shuttle. NRCs were infected either by adenovirus MRTFA-GFP alone at 50 MOI or by both adenovirus MRTFA-GFP and Life-Act-RFP. Cells were starved in serum-deprived DMEM medium for 24 h before the treatment. The immunofluorescent staining of α-cardiac actin (red) and fluorescent image of MRTFA-GFP (green), (**A**) basal conditions (serum free); (**B**) treatment with Ang II (100 nM). (**C**) The fluorescent image of MRTFA-GFP (green) and Life-Ac-RFP (red) with Ang II (100 nM) treatment. The white arrows indicate F-actin at 180 min. The immunofluorescent staining of α-cardiac actin (red) and fluorescent image of MRTFA-GFP (green), (**D**) treatment with Y-27632 (10 µM); (**E**) treatment with Ang II (100 nM) + Y-27632 (10 µM). Scale bar: 5 or 15 µm.

**Figure 5 ijms-23-07394-f005:**
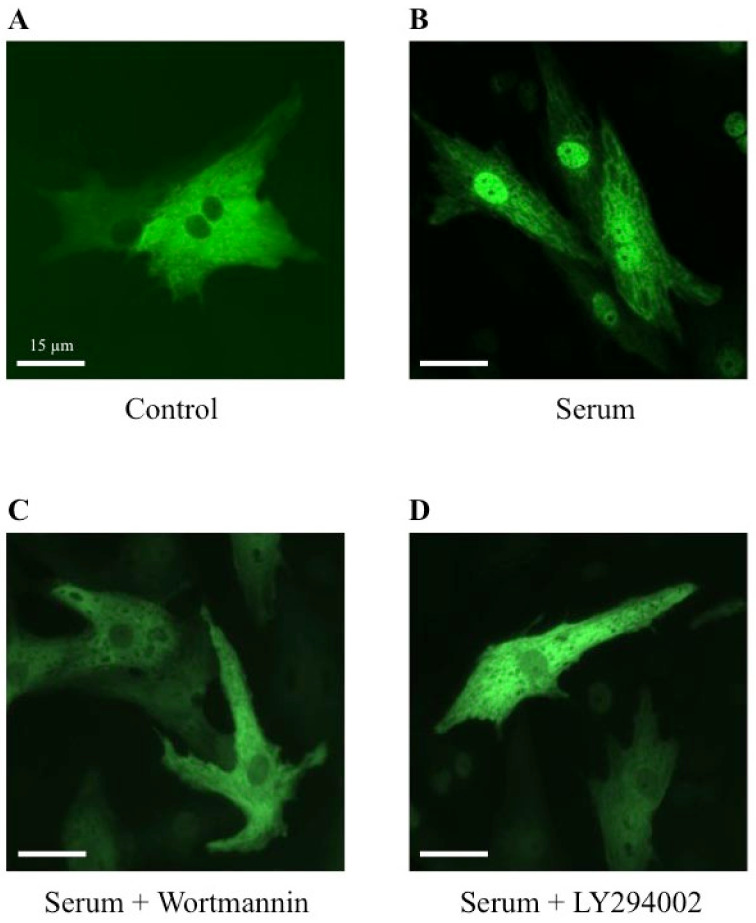
Effects of wortmannin and LY294002, two PI3K inhibitors, on the MRTFA-GFP shuttle induced by serum. NRCs were infected by adenovirus MRTFA-GFP at 50 MOI, starved in serum-deprived DMEM medium for 24 h before stress induction of 60 min. Fluorescence image of MRTFA-GFP, (**A**) basal conditions (serum free); (**B**) treatment with serum (20%); (**C**) treatment with serum (20%) + wortmannin (100 nM); (**D**) treatment with serum (20%) + LY294002 (30 µM). Scale bar: 15 µm. These data are representative of 3 independent experiments.

**Figure 6 ijms-23-07394-f006:**
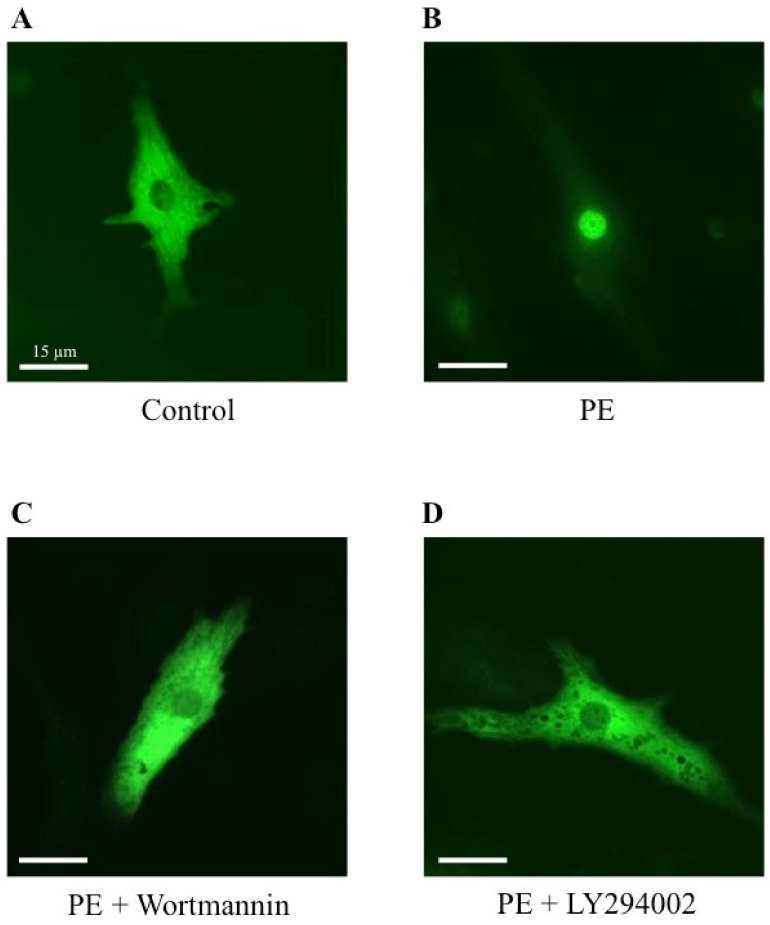
Effects of wortmannin and LY294002, two PI3K inhibitors, on the MRTFA-GFP shuttle induced by PE. NRCs were infected by adenovirus MRTFA-GFP at 50 MOI, starved in serum-deprived DMEM medium for 24 h before stress induction of 180 min. Fluorescence image of MRTFA-GFP, (**A**) basal conditions (serum free); (**B**) treatment with PE (100 nM); (**C**) treatment with PE (100 nM) + wortmannin (100 nM); (**D**) treatment with PE (100 nM) + LY294002 (30 µM). Scale bar: 15 µm. These data are representative of 3 independent experiments.

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
