# Peer review of "Interrelation between α-Cardiac Actin Treadmilling and Myocardin-Related Transcription Factor-A Nuclear Shuttling in Cardiomyocytes"

_ijms, 2022, doi:10.3390/ijms23137394_

Round 1

Reviewer 1 Report

Authors addressed previous concerns. Findings are interesting and merit publication.

Reviewer 2 Report

I have no further concerns

This manuscript is a resubmission of an earlier submission. The following is a list of the peer review reports and author responses from that submission.

Round 1

Reviewer 1 Report

In this study authors showed, that the nucleo-cytoplasmic shuttle of MRTFA is linked to filamentous actin dynamics in primary cultures of cardiomyocytes. Moreover, they provide evidence that MRTFA transport is dependent of the RhoA/ROCK signaling pathway, implying that actin signaling controlling filament dynamics is involved in this mechanism.

This study is interesting and well conducted. Authors used accurate visualizing techniques. Paper is well written.

The involvement of actin dynamics and implication of Rho/ROCK signaling in the nucleo-cytoplasmic transport mechanism of MRTFA is an important finding.

To my view however, additional experimental evidence, further supporting this conclusion, would considerably upgrade the impact of this study. To this end, some experiments are suggested below that may be considered by the authors.

  1. Indeed, authors may use actin microfilament drugs to mimic the effect of the ROCK inhibitor to MRTFA transport.
  2. Moreover, it would be interesting, by using appropriate inhibitors, to check the possible involvement of the PI-3K signaling in this process, since this signaling cascade represents an additional crucial signaling pathway regulating actin dynamics in various cells.

Reviewer 2 Report

Gorey et al. analyzed the translocation of GFP-MRTF into the nuclei of rat cardiomyocytes in response to stimulation with serum or with hormones. Furthermore, they analyzed the effect of the Rho-inhibitor Y-27632 in this process.

The fact that MRTF translocates into the nucleus after depletion of G-actin (due to formation of F-actin) is known for years. Also, the involvement of the Rho-formin axis is well-described. The authors reproduced the findings obtained from experiments with mouse fibroblasts with rat cardiomyocytes. This is interesting as a part of a study but does not have the potential to be published independently.  Thus, the authors are encouraged to include these findings in a study that aims to elucidate a new mechanism.